# Tracheal External Support Using a Parallel Loop Line Prosthesis for Tracheal Stent Fracture in a Dog: A Case Report

**DOI:** 10.3390/ani16020171

**Published:** 2026-01-07

**Authors:** Tomohiro Yoshida, Ryou Tanaka, Kazuyuki Terai, Aki Takeuchi, Akari Hatanaka, Daisuke Ito, Takashi Tanaka

**Affiliations:** 1Veterinary Teaching Hospital, Tokyo University of Agriculture and Technology, Fuchu 183-8509, Tokyo, Japan; fy3391@go.tuat.ac.jp (T.Y.); fu0253@go.tuat.ac.jp (R.T.); fq9913@go.tuat.ac.jp (K.T.); fv5028@go.tuat.ac.jp (A.T.); s257982z@st.go.tuat.ac.jp (A.H.); 2Department of Small Animal Clinical Sciences, Texas A&M University, College Station, TX 77843, USA; dice-k.ito@tamu.edu

**Keywords:** canine, external tracheal support, surgical complication management, laryngeal ultrasonography, extraluminal tracheal prosthesis (ELP), endoluminal tracheal stent (ELS), laryngeal paralysis, revision surgery, upper airway obstruction, postoperative complications

## Abstract

Tracheal collapse is a common respiratory disease in small dogs, and endoluminal tracheal stents are often used when medical treatment is not enough. However, stent fracture is a serious complication, and the best way to manage it is not well established. In this case, a dog developed severe breathing problems after its tracheal stent fractured. Instead of placing another stent, a parallel loop line prosthesis was used to support the trachea from the outside. This method restored the normal tracheal shape and maintained airway stability for several months. This case suggests that parallel loop line prosthesis may be a useful surgical option for dogs with fractured tracheal stents.

## 1. Introduction

Tracheal collapse is a progressive chronic respiratory disease commonly observed in small dogs [1,2]. Initial treatment involves medical management, such as weight control and administration of cough suppressants, which has been reported to improve clinical symptoms in over 70% of cases [1]. In the acutely collapsed patient or in dogs for which appropriate medical management is unsuccessful, surgical treatment may be considered [1]. Common surgical options for tracheal collapse are endoluminal tracheal stents (ELS) and extraluminal tracheal prostheses (ELP) [1,3,4].

ELS have many advantages, such as potential use in extra-thoracic and intra-thoracic tracheal collapse, minimally invasive, rapidity of insertion [5]. However, complications such as stent fracture, migration, granulation tissue formation, and chronic cough have been reported [5,6,7]. ELP are primarily used for extra-thoracic tracheal collapse [8]. In addition to commercially available devices, homemade methods using syringe sleeves have also been reported [9]. Prosthesis shapes include types with multiple ring-shaped elements arranged singly or continuously, and types with a single, continuous band-like structure, such as PLLP [9,10]. They are considered to be stable long-term and cause less mucosal irritation, resulting in fewer cases of granulation tissue formation and chronic cough [9,10,11]. However, complications such as infection, blood flow impairment, and laryngeal paralysis remain problematic [1,12].

In recent years, ELS are increasingly selected due to their simplicity of placement and rapid efficacy; however, stent fracture remains a significant complication [10,13]. Clear treatment guidelines for managing stent rupture have not been established. Reported approaches include re-stenting, stent removal followed by tracheal anastomosis, and even the additional placement of an ELP [5,6,12,13,14,15]. However, only two cases have been reported where an ELP was used to manage a ruptured endotracheal stent, and the detailed techniques and clinical course are not sufficiently documented [13,14].

The purpose of this case report is to describe the clinical course of canine tracheal stent fracture treated with external tracheal reinforcement using a PLLP with a continuous structure, and to discuss the potential utility, technical considerations, and associated postoperative complications, including laryngeal paralysis.

## 2. Case Description

A 4-year-old, 2.1 kg, castrated male Yorkshire Terrier was presented to the authors’ medical facility for further examination of tracheal collapse. For approximately two weeks before presentation, the dog had exhibited intermittent upper airway noise and episodic cyanosis. On initial examination, the dog had increased upper respiratory sounds. Physical examination revealed a body temperature of 38.2 °C, with a heart rate of 112 beats per minute and a respiratory rate of 42 breaths per minute, and a body condition score of 5/9.

Blood samples for complete blood count and serum biochemical analyses were collected via venipuncture of the lateral saphenous vein. Complete blood count (IDEXX ProCyte Dx; IDEXX Laboratories, Inc., Westbrook, ME, USA) was normal. Serum biochemical analyses (DRI-CHEM NX700; FUJIFILM Corporation, Tokyo, Japan) revealed mildly elevated alanine aminotransferase (ALT) (185 U/L; reference interval (RI) 17–78), aspartate aminotransferase (AST) (149 U/L; RI 17–44) and alkaline phosphatase (ALP) (136 U/L; RI 0–89). The results, including reference intervals, are summarized in Table 1.

Thoracic and cervical radiography revealed severe narrowing of the trachea from the caudal aspect of the fifth cervical to the cranial aspect of the seventh cervical vertebra (Figure 1). The heart, pulmonary vessels, and pulmonary parenchyma appeared within normal limits. Fluoroscopy revealed persistent narrowing during both inspiration and expiration, leading to a diagnosis of static tracheal collapse. The patient exhibited cyanosis during the fluoroscopy, but this improved with oxygen administration (FiO_2_ 0.4).

It was determined that medical treatment would not provide sufficient improvement and that surgical treatment was necessary. After consultation with the owner, ELS placement was chosen.

Subcutaneous injection of Atropine Sulfate (Atropine Sulfate Injection 0.5 mg; Mitsubishi Tanabe Pharma Co., Osaka, Japan, 0.05 mg/kg) and intravenous injections of Butorphanol (Vetorphale 5 mg; Meiji Animal Health Co., Ltd., Tokyo, Japan, 0.2 mg/kg) were used as pre-medications. Cefazolin Sodium (Cefazolin Sodium injection 1 mg; Nichi-Iko Pharmaceutical Co., Ltd., Toyama, Japan, 20 mg/kg) was administered intravenously as the perioperative antibiotic. General anesthesia was induced with Propofol (Propofol intravenous injection 1%; Fresenius Kabi, Tokyo, Japan, 3 mg/kg IV). After endotracheal intubation, general anesthesia was maintained with a mixture of Isoflurane (Isoflurane for animal use; MSD Animal Health, Rahway, NJ, USA) and oxygen.

Based on Computed Tomography (CT), the stent diameter was chosen to be 10–20% larger than the maximum diameter of the caudal trachea without collapse [14,15] (Figure 2). Although the stent length covered the entire trachea, it was selected so that the cranial end would not contact the larynx and the caudal end would not contact the tracheal carina [6,8,15]. A 10 mm diameter, 80 mm long, a Nitinol cross-and-hook braided self-expanding metal stent (Fauna Stent; M.I. Tech Co., Ltd., Seoul, Republic of Korea) was selected.

Following CT, bronchoscopy was performed to confirm the tracheal collapse in the caudal neck region. The stent was then placed according to the manual (Figure 3). After placement, bronchoscopy and fluoroscopy confirmed adequate expansion of the previously collapsed area. After extubation, the dog’s respiratory status was stable. Breathing remained stable the following day, and the dog was discharged. Follow-up examinations at 2 and 4 weeks postoperatively revealed only mild coughing (1–3 times a day) during excitement. Radiography showed no stent fracture or displacement. Subsequent regular examinations at 3, 6, and 11 months postoperatively showed good respiratory status with no abnormalities other than occasional coughing.

At one year and one month after ELS placement, the patient was re-presented to the hospital with increased coughing frequency and worsening respiratory sounds. Physical examination revealed inspiratory labored breathing and coughing. Radiography and fluoroscopy revealed that the stent had fractured at the tracheal collapse site identified at the initial presented (Figure 4A,B). Additionally, tracheal collapse was observed at the cranial end of the stent. Treatment options considered included placing a new tracheal stent or placing a PLLP. The owner requested PLLP placement, which was performed the following day.

The procedure for preparing the PLLP was as follows. The PLLP were prepared from a commercially available 0.75-mm-thick optical fibers (Optical Fiber Diameter 0.75 mm Length 2 m 10 fibers; UZIPAL Co., Shanghai, China). Based on the tracheal diameter estimated by CT, a Terumo Syringe 2.5 mL (Disposable syringe; TERUMO Co., Tokyo, Japan) was used as a mold. Nails (diameter 1.2 mm, length 16 mm) were inserted in a single row approximately 5 mm apart [16]. The optical fiber was then wound around the nails in a zigzag pattern for fixation. The mold was heated in boiling water for 1 min and then rapidly cooled in cold water. After trimming excess material and drying, the device underwent gas sterilization [16].

Subcutaneous injection of atropine sulfate (0.05 mg/kg) and intravenous injections of Fentanyl (Fentanyl Injection 0.25 mg; Daiichi Sankyo Propharma Co., Ltd., Tokyo, Japan, 5 mcg/kg) were used as pre-medications. Cefazolin Sodium (20 mg/kg) was administered intravenously as the perioperative antibiotic. General anesthesia was induced with Propofol (4 mg/kg IV). After endotracheal intubation, general anesthesia was maintained with a mixture of isoflurane and oxygen. Intraoperative analgesia was provided via a constant rate infusion of Fentanyl (2.5 to 10 mcg/kg/h).

Bronchoscopy revealed that the tracheal stent was entirely covered by granulation tissue, obscuring its structure. At the site of fractured, a portion of the stent protruded into the tracheal lumen (Figure 4D). Additionally, the ventral aspect of the tracheal cartilage was found to be invaginated toward the lumen, resulting in flattening of the tracheal lumen.

A ventral midline approach to the cervical trachea was performed. Dissection was performed while preserving the tracheal blood vessels as much as possible, and the recurrent laryngeal nerves were identified under direct visualization. The fractured area of the stent showed deformation of the cartilage, which had collapsed into the tracheal lumen and flattened (Figure 4C).

To avoid injury to the recurrent laryngeal nerve, umbilical cord tape was applied to the cranial and caudal sides of the planned PLLP placement site, allowing traction only on the trachea. The PLLP was positioned to cover the damaged area and the cephalic end of the trachea. To prioritize repair of the ventral deformation, the ring opening was oriented dorsally.

A 4-0 Polypropylene Monofilament suture (PROLENE 17 mm 3/8c taperpoint; Johnson & Johnson, Tokyo, Japan) was used for fixation. The ligature was performed by threading the suture through the endotracheal stent as well, thereby pulling the damaged stent while reshaping the tracheal contour. On the ventral side, 3–4 ligations were placed at the fractured site and 2–3 at the non-lesional site. Additionally, one fixation/support suture was added to each ventrolateral side to enhance operability. Pulling the support suture mildly rotated the trachea, allowing ligation of the dorsal loop and reshaping the trachea into a circular form (Figure 4G).

After all sutures were placed, sterile saline was instilled into the trachea to confirm no air leakage from the suture sites. Bronchoscopy revealed the stent protruding into the tracheal lumen had disappeared, and the previously flattened tracheal lumen had clearly improved to a circular shape (Figure 4H).

Postoperatively, respiration remained stable, and the patient was managed in the intensive care unit. While in the intensive care unit, medical management consisted of supplemental oxygen (FiO_2_, 0.4), intravenous Cefazolin Sodium (20 mg/kg, q 12 h), and Fentanyl (2.5 mcg/kg/h) administered via CRI for analgesia. After discontinuing the fentanyl CRI, Butorphanol (0.2 mg/kg, IV, q 8 h) was administered to also suppress coughing. During hospitalization, mild coughing was only observed during periods of excitement. The patient was discharged on postoperative day 6.

Seven days after discharge, the dog returned with inspiratory effort, cyanosis, and loss of vocalization. Radiography revealed no obvious abnormalities. Laryngeal ultrasound confirmed absence of motion of the cuneiform processes and caudal displacement of the whole larynx, leading to a diagnosis of laryngeal paralysis (Figure 5). Given that breathing was stable at rest and the owner declined anesthesia, medical management was chosen.

Wet breath sounds were heard during respiration, often followed by labored breathing. Nebulizer therapy at home and L-Carbocisteine (Mucodyne tablets 250 mg; Kyorin Pharmaceutical Co., Ltd., Tokyo, Japan, 15 mg/kg, q 12 h) were prescribed to reduce secretion viscosity. This reduced breath sounds and improved labored breathing, but inspiratory effort recurred during excitement. Therefore, Acepromazine Maleate (ACEPRON Injection; Panav Bio-Tech Pvt Ltd., New Delhi, India, 0.05–0.3 mg/kg, PO) was added for sedation. Initially, 0.05 mg/kg was administered 2–3 times daily, with doses increased to 0.1–0.3 mg/kg when significant agitation was anticipated.

By approximately day 4 after starting medication, the respiratory status stabilized, and inspiratory effort decreased to about once per day. At the 4-week postoperative follow-up, nebulizer stimulation at home induced excitement, so nebulizer therapy was discontinued (Figure 6).

Although vocalization recovered after 3 months, laryngeal ultrasound examination showed no improvement in arytenoid cartilage movement, leading to the conclusion that laryngeal paralysis persisted.

Five months after PLLP placement, the dog was presented to the hospital with inspiratory labor and cyanosis. Physical examination revealed inspiratory effort was noted, along with abnormal breath sounds suggestive of laryngeal paralysis. Radiography revealed no evidence of stent failure or displacement. After intravenous administration of Alfaxalone (Alfaxan Multidose; Meiji Animal Health Co., Ltd., Tokyo, Japan, 2 mg/kg) for laryngeal examination, no abducent movement of the arytenoid cartilages was observed during inspiration, leading to a diagnosis of laryngeal paralysis. Bronchoscopy showed no abnormalities within the trachea.

As a treatment option, unilateral arytenoid cartilage lateralization was recommended and performed with the owner’s consent, with concurrent bronchoscopy performed at the time of surgery (Figure 7) Postoperatively, the respiratory status was temporarily stable, but the patient developed aspiration pneumonia on the second postoperative day and died.

## 3. Discussion

Endotracheal stent fracture is a major complication in the management of canine tracheal collapse, with a reported incidence ranging from 9.1% to 50% following stent placement, underscoring its considerable clinical significance [7,8,14,15,17]. Identifying the underlying causes of stent fracture and establishing appropriate strategies for its prevention and treatment are therefore of paramount importance. In the present dog, the stent fracture was considered to have resulted from localized stress concentration caused by repeated mechanical forces associated with chronic coughing, combined with the possibility of suboptimal stent length selection. Previous reports similarly indicate that body weight gain, chronic airway irritation and inappropriate stent sizing, length can precipitate stent fracture, which is consistent with the persistent postoperative coughing observed in this case [5,6,7,8,12,14,15,17].

With regard to stent positioning, the cranial end of the stent was located just caudal to the fourth cervical vertebra, leaving a short distance (approximately 5 mm) between the stent and the cranial margin of the collapsed segment at the level of the fifth cervical vertebra. This short distance likely concentrated excessive bending stress in this region, contributing to the fractured. Although some literature recommends leaving a margin of approximately 10 mm from the collapsed segment, a definitive standard has not been established [1,3,6]. The outcome of this case underscores the importance of carefully determining both stent length and placement to minimize the risk of fracture. In the dog, after fracture of the endotracheal stent, external tracheal support using an extraluminal PLLP prosthesis was selected instead of re-stenting with an additional endotracheal stent or applying a single ring-type extraluminal prosthesis. Re-stenting was avoided because chronic coughing posed a high risk of recurrent fracture of any newly placed stent, and because the placement of an additional stent could further expand the fractured original stent, allowing frayed stent fragments to migrate into and potentially damage the surrounding soft tissues.

The PLLP was considered advantageous because its continuous structure provides broad and uniform support and traction to the tracheal wall, thereby minimizing localized stress concentration and effectively dispersing mechanical loads [4,16]. In contrast, single ring-type extraluminal prostheses offer only localized points of support, which were judged insufficient to achieve adequate reshaping in a case complicated by a fractured intraluminal stent [10,18]. Additionally, the PLLP allows suturing from multiple directions to the mesh-like stent, improving flexibility in fixation. This ability to achieve multipoint anchoring—difficult to accomplish with ring-type devices—was another important reason for selecting the PLLP in this case [10].

Beyond these theoretical advantages, surgical application of the PLLP was tailored to the specific morphological changes observed in the dog. PLLP is generally placed via a ventral approach with the looped portion positioned ventrally, which facilitates surgical exposure, improves operability, and enables uniform external traction of the membranous trachea [10]. However, in the present case, pronounced ventral stent damage and tracheal deformation were evident; thus, the non-looped side was intentionally placed ventrally to prioritize reshaping. This orientation provided better access for multidirectional suturing and allowed restoration of a tracheal contour closer to its original shape.

To assess the durability of this external tracheal support over time, bronchoscopic findings obtained during subsequent unilateral arytenoid cartilage lateralization were reviewed. Despite persistent negative intraluminal pressure associated with laryngeal paralysis for five months, the tracheal lumen remained well maintained in a circular configuration. No progression of collapse at the cranial end of the stent or additional structural failure was observed. To date, no studies have evaluated the relative utility of ring-type prostheses versus PLLP for external reinforcement following stent failure. The present findings suggest that PLLP can serve as an effective complementary structure for stabilizing and reshaping a damaged tracheal stent.

Despite the maintenance of tracheal patency after PLLP placement, the postoperative course was complicated by the development of laryngeal paralysis. Complications of laryngeal paralysis following ELP placement have been reported to be 11–21% [9,10]. Causes include recurrent laryngeal nerve injury, postoperative inflammation or edema, or chronic compression by the ELP [9,19]. In the dog, the nerve was visualized intraoperatively and no obvious injury was observed. Furthermore, no signs of paralysis were noted immediately postoperatively. Therefore, the paralysis was considered a transient functional impairment associated with postoperative inflammatory changes.

The owner declined additional anesthesia, so conservative management was chosen for the dog. Specifically, this involved reducing secretion viscosity using a nebulizer and L-Carbocisteine, suppressing agitation with Acepromazine Maleate, and alleviating excessive negative inspiratory pressure [20,21]. These measures resulted in partial stabilization of the respiratory condition. However, although vocalization partially recovered after 3 months, it was actually hoarse. Ultrasound examination also failed to confirm improvement in arytenoid abduction, suggesting that the laryngeal paralysis had not fundamentally improved. Subsequently, lateralization of the arytenoid cartilage was performed due to worsening respiratory status, but the dog developed aspiration pneumonia postoperatively and died.

Despite the unfavorable final outcome, this case suggests that conservative treatment may offer short-term symptomatic relief in selected dogs with suspected reversible laryngeal paralysis—particularly in individuals whose temperament maintains adequate rest. Nonetheless, this approach requires careful monitoring, as early identification of cases unlikely to improve is crucial to determine the appropriate timing for surgical intervention. In the dog, laryngeal ultrasound was used to assess suspected laryngeal paralysis. Although definitive diagnosis in dogs is typically achieved via laryngoscopy under mild sedation, the owner declined sedation; therefore, un-sedated ultrasonography was selected as the initial diagnostic modality [22]. Rudorf et al. (2001) reported that laryngeal ultrasound can accurately detect the presence and laterality of laryngeal paralysis [23]. In the present case, ultrasonography without anesthesia demonstrated an absence of cuneiform process motion and caudal displacement of the entire larynx. These findings were consistent with the characteristic ultrasonographic features described in previous paralysis cases [23]. Subsequent direct visualization during unilateral arytenoid lateralization confirmed the lack of arytenoid abduction during inspiration, thereby validating the ultrasound diagnosis. Laryngeal ultrasound offers several important advantages: it can be performed safely without sedation, poses minimal risk even in patients with severe respiratory distress, and allows real-time assessment of laryngeal motion [23]. It is particularly valuable when sedation is contraindicated or declined, as in the present case. Thus, this dog illustrates that laryngeal ultrasound provides sufficient diagnostic value for identifying laryngeal paralysis, even when standard sedated laryngoscopy cannot be performed. Overall, the dog illustrates the potential utility of PLLP as a supportive structure for managing tracheal stent fractures and highlights both the challenges and possible roles of conservative management in cases of postoperative laryngeal paralysis. The findings emphasize the need for further case accumulation to establish optimal criteria for device selection, appropriate surgical indications, and long-term outcomes.

### Limitations

This report describes a single clinical case; therefore, the findings cannot be generalized, and causal relationships between PLLP placement and clinical outcomes cannot be definitively established. In particular, the long-term durability of PLLP as external tracheal support and its comparative efficacy relative to other salvage strategies, such as repeat intraluminal stenting or ring-type extraluminal prostheses, remain unclear.

In addition, the development of laryngeal paralysis in this dog introduces potential confounding factors that may have influenced postoperative respiratory dynamics and overall outcome. Although a direct causal relationship between PLLP placement and laryngeal paralysis could not be confirmed, its presence limits interpretation of long-term clinical results.

Further accumulation of cases and prospective studies are required to better define appropriate indications, surgical techniques, and long-term outcomes of PLLP placement for managing tracheal stent fracture in dogs.

## 4. Conclusions

In the present case, placement of a PLLP was associated with restoration and maintenance of a near-normal tracheal lumen following ELS fracture in a dog. External tracheal support using PLLP contributed to stabilization of the tracheal contour and prevention of further luminal deformation during the observation period.

As this report describes a single case, these findings should be interpreted cautiously. Nevertheless, this case suggests that PLLP placement may represent a potential surgical approach for maintaining tracheal structure after stent fracture. Further accumulation of cases and systematic evaluation are required to define indications, refine surgical techniques, and assess long-term outcomes.

## Figures and Tables

**Figure 1 animals-16-00171-f001:**
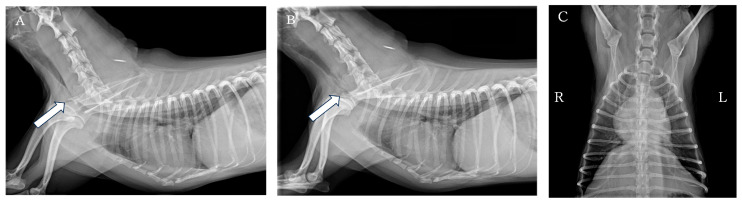
Cervicothoracic radiographs of the dog. (**A**) Right lateral view during inspiration. (**B**) Right lateral view during expiration. (**C**) Ventrodorsal view. The trachea was severely collapsed from the level of the caudal aspect of the fifth cervical to the cranial aspect of the seventh cervical. Arrows: the region of tracheal collapse.

**Figure 2 animals-16-00171-f002:**
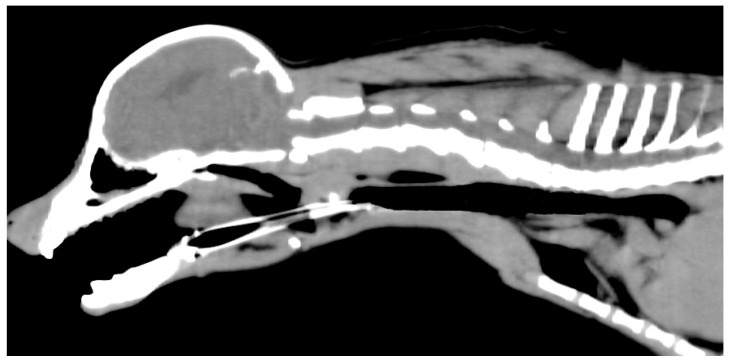
Computed tomography images of the dog. Based on these images, the maximum diameter of the caudal trachea without collapse was estimated to be approximately 9 mm. The stent length was selected so that the cranial end would not contact the larynx and the caudal end would not contact the tracheal carina.

**Figure 3 animals-16-00171-f003:**
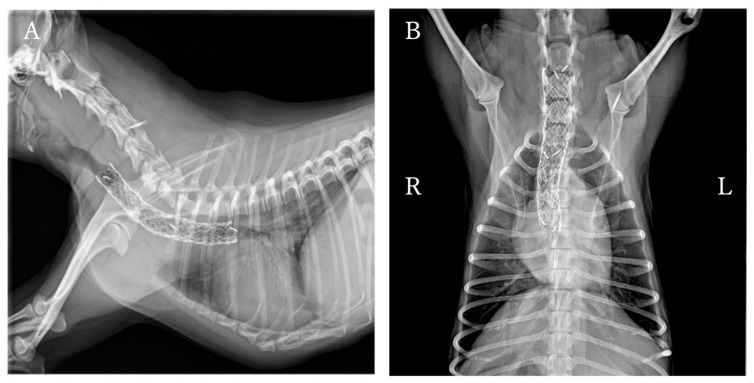
Cervicothoracic radiographs of the dog in Figure 1 after placement of an endoluminal tracheal stent. (**A**) Right lateral view. (**B**) Ventrodorsal view. The stent spans and expands the area of tracheal collapse.

**Figure 4 animals-16-00171-f004:**
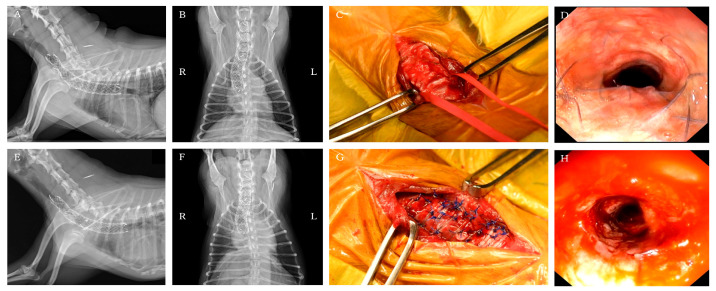
Pre- and post-placement findings of the PLLP used to manage tracheal stent fracture. (**A**–**D**) Pre-PLLP placement images. (**E**–**H**) Post-PLLP placement images. (**A**) Cervicothoracic radiographs in the right lateral view showing complete fracture of the endoluminal tracheal stent, with associated tracheal narrowing at the fracture site and cranial to the stent. (**B**) Cervicothoracic radiographs in the ventrodorsal view. (**C**) Cervical view demonstrating protrusion of a fractured stent segment into the tracheal lumen. Ventral tracheal cartilage invagination was also present, resulting in luminal flattening. (**D**) Bronchoscopy showing marked deformation of the tracheal lumen at the fracture site. (**E**) Cervicothoracic radiographs in the right lateral view showing successful reshaping and stabilization of the intraluminal stent. (**F**) Cervicothoracic radiographs in the ventrodorsal view. (**G**) Cervical view demonstrating restoration of a near-normal tracheal contour. (**H**) Bronchoscopy confirming improvement in tracheal luminal shape.

**Figure 5 animals-16-00171-f005:**
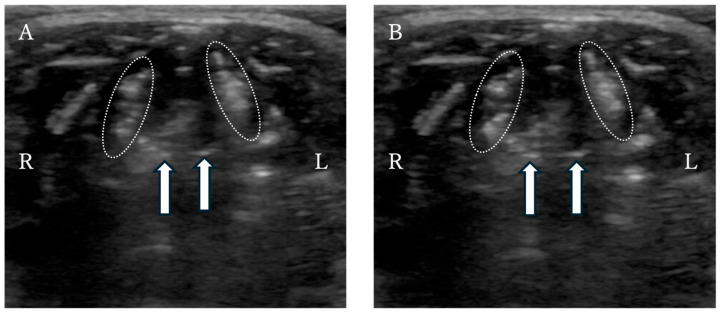
Laryngeal ultrasound of the dog revealed a lack of cuneiform process movement during (**A**) inspiration and (**B**) expiration. The dotted outline: the arytenoid cartilage. Arrows: cuneiform processes of the arytenoid cartilages.

**Figure 6 animals-16-00171-f006:**
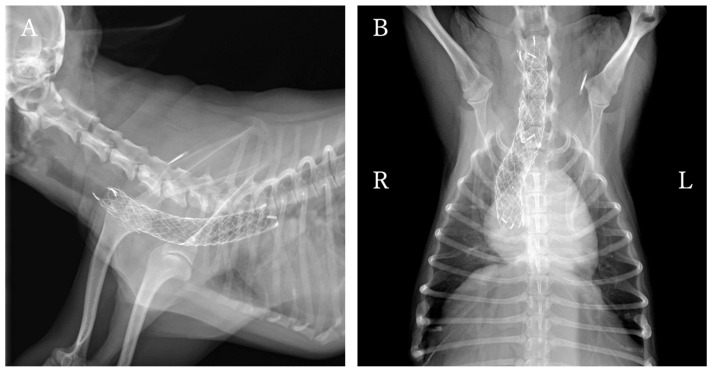
Cervicothoracic radiographic view of the dog obtained during postoperative follow-up after PLLP placement. (**A**) Right lateral view. (**B**) Ventrodorsal view. No stent deformation or tracheal narrowing at the fracture site or cranial to the stent was observed.

**Figure 7 animals-16-00171-f007:**
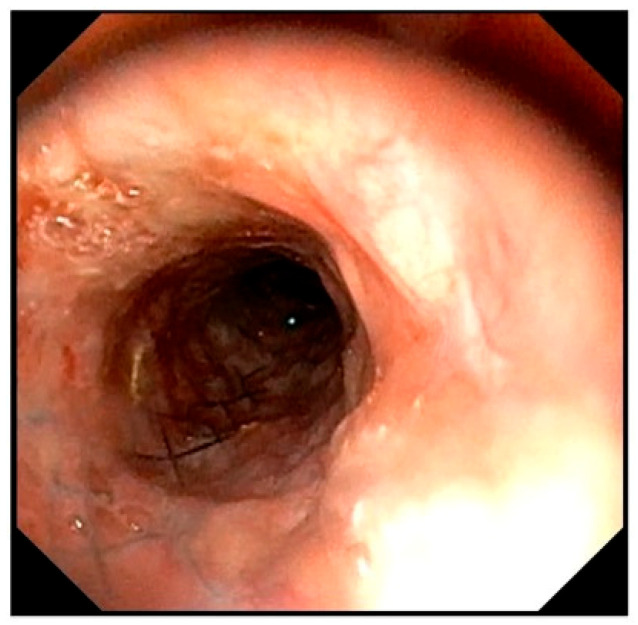
Bronchoscopic image obtained during unilateral arytenoid cartilage lateralization. The tracheal lumen maintains a circular shape.

**Table 1 animals-16-00171-t001:** Blood examination findings at initial presentation.

Parameter	Unit	Value	Reference Interval
RBC	M/µL	7.70	5.65–8.87
Hct	%	49.8	37.3–61.7
WBC	K/µL	9.73	5.05–16.76
Plt	K/µL	318	148–484
ALT	U/L	185	17–78
AST	U/L	149	14–44
ALP	U/L	136	0–89
TBil	mg/dL	0.1	<0.5
BUN	mg/dL	12.6	9.2–29.2
Cre	mg/dL	0.4	0.4–1.4
TP	g/dL	6.4	5.0–7.2
Alb	g/dL	3.1	2.6–4.0
Glu	mg/dL	121	75–128
TCho	mg/dL	185	115–337
Na	mEq/L	150	141–152
K	mEq/L	4.3	3.8–5.0
Cl	mEq/L	102	102–11
CRP	mg/dL	0.6	<0.7

RBC, red blood cell count; Hct, hematocrit; WBC, white blood cell count; Plt, platelet count; ALT, alanine aminotransferase; AST, aspartate aminotransferase; ALP, alkaline phosphatase; TBil, total bilirubin; BUN, blood urea nitrogen; Cre, creatinine; TP, total protein; Alb, albumin; Glu, glucose; TCho, total cholesterol; Na, sodium; K, potassium; Cl, chloride; CRP, C-reactive protein.

## Data Availability

The original contributions presented in this study are included in the article/Appendix A. Further inquiries can be directed to the corresponding authors.

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
