# Peer review of "Tracheal External Support Using a Polypropylene Linear Prosthesis for Tracheal Stent Fracture in a Dog: A Case Report"

_animals, 2026, doi:10.3390/ani16020171_

Round 1
Reviewer 1 Report
Comments and Suggestions for Authors
First of all, I found this manuscript to be very well written, clear and clinically valuable. The case is presented in a structured manner, the surgical technique is desvribed in excellent detail, and the discussion is well supported by the literatures. The use of PLLP as an external support for managing trahceal stent fractures is both interesting and innovative. However, I have a few minor question for clarification
1- Although tracheal patency was successfully maintained after PLLP placement, the dog ultimately developed persistent laryngeal paralysis (LP) and died following aspiration pneumonia. A more explicit discussion of how this unfavorable outcome affects the overall interpretation of PLLP as a salvage option would strengthen the manuscript.
2- The causal relationship beetween PLP placement and the development of LP remains unclear. Further clarification on whether this complication is considered coincidentaÅŸ, inflammatory or mechanically related would be helpful
3- As this a single case... The conclusion describing PLLP as a surgion option may benefit fom slightly more cautious wording to emphasize the hypothesis-generating nature of the findings in my opinion.
I would like to say, this is strong and well prepared case report. and adrrresing the point would further impruve ist scientific balance.
Reviewer 2 Report
Comments and Suggestions for Authors
SUMMARY
This manuscript describes a complex clinical case of a dog with tracheal collapse treated with an endoluminal tracheal stent that subsequently developed multiple complications. The manner in which the authors addressed these complications is of interest and contributes valuable insight into the current knowledge of this pathology and its associated complications. Additionally, the manuscript presents a novel diagnostic approach for laryngeal paralysis.
Overall, the study is methodologically sound, and the discussion is clearly presented. However, several minor revisions are required:
-
Line 230: The description of cefazolin administration is unclear. At the beginning of the sentence, cefazolin is reported to have been administered every 12 hours; however, it is later described as being administered “continuously.” Please clarify this inconsistency.
-
Lines 385–396: The conclusions are not clearly distinguished from the objectives and appear to be intermixed. For example, the statement “Tracheal stent fracture is a serious and potentially life-threatening complication following endoluminal stent placement for canine tracheal collapse, and optimal management strategies have not yet been established” does not constitute a conclusion of the present study. Please revise this section to clearly and explicitly state the conclusions derived from the reported case.
-
Discussion: Please include a section addressing the limitations of the study.
ADDITIONAL FEEDBACK
Main research question:
The primary question addressed in this study is the appropriate management and treatment of complications following the placement of a tracheal stent for the treatment of canine tracheal collapse.
Relevance of the topic:
The topic is highly relevant to the field. While the placement of tracheal stents is well described in the literature, the management of potential complications is less extensively documented. This case contributes valuable information and enriches the scientific literature.
Contribution to the subject area:
The manuscript provides insight into the management of complications in a complex clinical case, offering practical guidance that complements existing publications on tracheal stenting.
Consistency of conclusions with evidence and main question:
Yes, the conclusions are consistent with the evidence and arguments presented in the manuscript and appropriately address the primary research question.
Appropriateness of references:
Yes, the references cited are appropriate and relevant to the topic.
Reviewer 3 Report
Comments and Suggestions for Authors
Dear authors,
Some suggestions are pointed above, to improve the article quality.
- Title: consider include “case report” to the title; consider remove “PLLP”
- Abstract: consider presented it with one paragraph;
- Keywords: “dog”, “tracheal stent fracture” and “Polypropylene linear prosthesis (PLLP)” are presented at title, consider change it;
- Line 64-67: it is presented part of the case, which can be presented at case description; consider present a revision regarding stent complication, including causes, frequency, between other important data related to the problem;
- Figure 1, 2 and 3: the radiograph exam is valid just when two perpendicular views are evaluated; consider present the missing exposition; in the figure title, consider use “cervicothoracic radiographs”;
- Line 185-192: consider present the procedure describe in this section;
- Line 202: regarding Figure 3C, consider show the lesion area of bleeding using an arrow; consider highlighting the important image details in the same way;
- Line 220: abdominal side? Consider reviewing the anatomical term;
- Line 337: “an effective complementary”; considering the proposed technique was used in just one patient, I consider dangerous this affirmation; consider evaluated it, considering in one case no complication rate was possible to evaluate;
- Line 355: “conservative treatment” is it related to laryngeal paralysis? Consider clear it;
The article brings as main objective “reports the clinical course of this rare case and discusses the usefulness and considerations of external reinforcement using a PLLP.” Although, observing the discussion, a good discussion related to laryngeal paralysis is presented, adequately; consider improve the objective regarding the complication of this serious disease presentation.
Reviewer 4 Report
Comments and Suggestions for Authors
General comments:
Although the topic is not entirely novel, the authors report an interesting case involving the use of a polypropylene linear prosthesis as an external tracheal support in a dog. The manuscript falls within the scope of the journal, and the writing style is generally adequate.
Title:
The title is short, concise, and appropriate.
Abstract and keywords:
The abstract is complete and informative. However, the keywords should differ from those already included in the title.
Case description:
The case description requires further clarification and expansion. Specifically:
-
Please describe how blood samples were collected for complete blood count and serum biochemical analyses.
-
The full terms should be provided at first mention for the abbreviations GOT, GPT, and ALP.
-
A table summarizing the results of the blood analyses should be included.
-
In Figure 1, the tracheal collapse should be clearly indicated using an arrow.
-
Computed tomography images should be provided.
-
Follow-up images should be included to document the postoperative outcome.
-
Images should not be included in the Discussion section.
Discussion:
The Discussion section should be improved, as the current text lacks coherence and logical flow between sentences.
Conclusion:
The conclusions are appropriate. The full term should be removed before the abbreviation “PLLP.”
References:
Please standardize the reference formatting, as journal titles are inconsistently abbreviated. If possible, more recent references and a greater number of citations should be added to strengthen the manuscript.
